# *Cdon* mutation and fetal alcohol converge on Nodal signaling in a mouse model of holoprosencephaly

**Mingi Hong[1], Annabel Christ[2], Anna Christa[2], Thomas E Willnow[2], Robert S Krauss[1]\***

[1]Department of Cell, Developmental, and Regenerative Biology, Icahn School of Medicine at Mount Sinai, New York, United States; [2]Max-Delbruck-Center for Molecular Medicine, Berlin, Germany

**Abstract** Holoprosencephaly (HPE), a defect in midline patterning of the forebrain and midface, arises ~1 in 250 conceptions. It is associated with predisposing mutations in the Nodal and Hedgehog (HH) pathways, with penetrance and expressivity graded by genetic and environmental modifiers, via poorly understood mechanisms. CDON is a multifunctional co-receptor, including for the HH pathway. In mice, *Cdon* mutation synergizes with fetal alcohol exposure, producing HPE phenotypes closely resembling those seen in humans. We report here that, unexpectedly, Nodal signaling is a major point of synergistic interaction between *Cdon* mutation and fetal alcohol. Window-of-sensitivity, genetic, and in vitro findings are consistent with a model whereby brief exposure of *Cdon* mutant embryos to ethanol during gastrulation transiently and partially inhibits Nodal pathway activity, with consequent effects on midline patterning. These results illuminate mechanisms of gene-environment interaction in a multifactorial model of a common birth defect.

**\*For correspondence:**
Robert.Krauss@mssm.edu

**Competing interests:** The authors declare that no competing interests exist.

## Introduction

Many common structural birth defects appear to arise from a complex and ill-defined combination of genetic and environmental factors (*Krauss and Hong, 2016*). The notion that environmental agents affect penetrance and expressivity of predisposing mutations is speculated to underlie many human congenital anomalies, but has been difficult to demonstrate. Animal models are therefore valuable, but these have been slow to emerge. Some recent successful examples are a zebrafish model of craniofacial defects (*Pdgfra* mutations and fetal alcohol), and mouse models of scoliosis (Notch pathway mutations plus hypoxia), facial clefting (*Pax3* mutations plus 2,3,7,8-tetrachlorodibenzodioxin (TCDD)), and holoprosencephaly (Hedgehog pathway mutations and fetal alcohol) (*Hong and Krauss, 2012*; *Kietzman et al., 2014*; *McCarthy et al., 2013*; *Sparrow et al., 2012*; *Zalc et al., 2015*).

Holoprosencephaly (HPE) is a common developmental defect in midline patterning of the forebrain and/or midface (*Muenke and Beachy, 2001*). HPE occurs approximately once per 250 human conceptions, with an associated in utero lethality of ~97% (*Leoncini et al., 2008*; *Shiota and Yamada, 2010*). HPE is characterized by an unbroken continuum of rostroventral midline anomalies that range from complete failure to partition the forebrain into hemispheres plus cyclopia, through progressively less severe defects including partially partitioned forebrain, single nostril, and midface hypoplasia (*Cohen, 2006*; *Krauss, 2007*; *Muenke and Beachy, 2001*).

Development of the rostroventral midline is initiated by signals derived from the prechordal plate (PCP) (*Kiecker and Niehrs, 2001*; *Marcucio et al., 2011*). The PCP produces Sonic hedgehog (SHH), initiating a sequence of events that progressively patterns the forebrain and midface (*Aoto et al., 2009*; *Cordero et al., 2004*; *Geng and Oliver, 2009*; *Kiecker and Niehrs, 2001*;

**eLife digest** A common birth defect known as holoprosencephaly affects how the brain and face of a fetus develop in the womb. In many cases, the condition is so severe that the fetus dies before, or shortly after, birth. Mutations in certain genes that control how the fetus develops are associated with holoprosencephaly. For example, mutations in components of the Hedgehog and Nodal signaling pathways, which transmit information that help cells to become specialized, increase the risk that a fetus will develop holoprosencephaly. Environmental factors, such as exposure to alcohol in the womb, are also thought to contribute to this condition.

A gene known as *Cdon* is a component of the Hedgehog signaling pathway. In 2012, a team of researchers reported that mice with a mutation in the *Cdon* gene exposed to alcohol in the womb develop symptoms similar to holoprosencephaly in humans. Here, Hong et al. – including some of the researchers involved in the previous work – set out to understand how *Cdon* and alcohol work together to cause holoprosencephaly in the mutant mice.

First, the team exposed pregnant mice to alcohol at different times during gestation to find out when their young were sensitive to developing holoprosencephaly. This showed that the young mice were most sensitive in early pregnancy when the Nodal pathway was active in their growing bodies. Further experiments found that alcohol and mutations in *Cdon* change Nodal signaling in cells. Together, these findings demonstrate that exposure to alcohol in the womb works together with the mutant form of *Cdon* via the Nodal signaling pathway, rather than the Hedgehog pathway, to cause holoprosencephaly in mice.

The causes of many common birth defects are complex and difficult to distinguish at the level of individual cases. The work of Hong et al. illuminates how multiple risk factors during pregnancy, which may not create any problems on their own, may work together to produce birth defects in the fetus. The findings also offer new ways to understand how exposure to alcohol in the womb affects the fetus. Ultimately, understanding how birth defects form could lead to new strategies to prevent them in the future.

*Rubenstein and Beachy, 1998*; *Zhang et al., 2006*). HH ligands trigger signaling by binding the primary receptor Patched1 (PTCH1). In the absence of ligand, PTCH1 constrains the activity of a second membrane protein, Smoothened (SMO). HH binding inhibits PTCH1 function, allowing SMO-dependent signals to activate GLI transcription factors and pathway-specific gene expression. HH ligand availability and signal reception are also regulated by a series of co-receptors and additional HH-binding proteins, including CDON, BOC, GAS1, and LRP2 (*Beachy et al., 2010*; *Kong et al., 2019*; *Lee et al., 2016*; *Petrov et al., 2017*; *Willnow and Christ, 2017*).

The PCP develops from the anterior primitive streak (APS) under the influence of Nodal pathway signaling (*Robertson, 2014*; *Schier, 2009*; *Shen, 2007*). Therefore, the Nodal pathway is developmentally upstream of the HH pathway in rostroventral midline patterning. Consistent with this conclusion, defective Nodal signaling at the primitive streak stage of development leads to defects in HH signaling and HPE (*Krauss, 2007*). Nodal is a TGFβ superfamily member and signals through a receptor complex including the type I and type II receptors, ALK4 and Activin receptor IIA/B (ActRIIA/B), and a GPI-linked co-receptor, TDGF1 (also called Cripto) (*Robertson, 2014*; *Schier, 2009*; *Shen, 2007*). Receptor activation results in phosphorylation of the transcription factors SMAD2 and SMAD3, and activation of genes required for PCP induction (*Robertson, 2014*; *Schier, 2009*; *Shen, 2007*). Among these genes are *Foxa2* and *Gsc* (*Ang and Rossant, 1994*; *Belo et al., 1998*; *Filosa et al., 1997*). Nodal signaling also induces expression of *Lefty1* and *Lefty2*, which encode secreted inhibitors of the pathway that bind to TDGF1 and Nodal itself (*Robertson, 2014*; *Schier, 2009*; *Shen, 2007*).

The etiology of HPE is complex, involving both genetic and environmental risk factors (*Addissie et al., 2020*; *Dubourg et al., 2018*; *Hong and Krauss, 2018*; *Johnson and Rasmussen, 2010*; *Krauss, 2007*; *National Birth Defects Prevention Study et al., 2010*; *Muenke and Beachy, 2001*; *NISC Comparative Sequencing Program et al., 2018*; *Roessler et al., 2018*; *Summers et al., 2018*). Heterozygous, loss-of-function mutations in components or regulators of the HH, Nodal, and FGF signaling pathways are associated with HPE (*NISC Comparative Sequencing*

*Program et al., 2018*; *Roessler et al., 2018*). Epidemiology of HPE is less advanced than genetic analyses, but among the environmental risk factors implicated is fetal alcohol exposure (*Abe et al., 2018*; *Cohen and Shiota, 2002*; *Croen et al., 2000*), though this is not always observed (*Addissie et al., 2020*). A full range of clinical phenotypes is seen in both sporadic and familial HPE (*Muenke and Beachy, 2001*; *Solomon et al., 2010*). Many mutation carriers in pedigrees lack clinical manifestation, and many apparently sporadic cases have inherited a mutation from a minimally affected parent (*Lacbawan et al., 2009*; *Solomon et al., 2012*). It is likely, therefore, that HPE-associated mutations are not sufficient to produce midline defects, but are the substrate on which a complex landscape of genetic and/or environmental modifiers act. Statistical analysis is consistent with a multifactorial, 'autosomal dominant with modifier' model, wherein the penetrance and expressivity of a heterozygous driver mutation is graded by the modifier landscape (*Dubourg et al., 2018*; *Hong and Krauss, 2018*; *Roessler et al., 2012*).

We have modeled this phenomenon in mice with high fidelity. *CDON* encodes a multi-functional co-receptor, including for the HH pathway, (*Lu and Krauss, 2010*; *Sanchez-Arrones et al., 2012*). *CDON* loss-of-function mutations have been identified in human HPE patients (*Bae et al., 2011*; *NISC Comparative Sequencing Program et al., 2018*), but such variants are relatively common in the human population. Additionally, a patient with a rare homozygous *CDON* mutation displayed retinal coloboma, a mild HPE-associated eye phenotype also seen in *Cdon*-/- mice (*Berkun et al., 2019*; *Pineda-Alvarez et al., 2011*; *Zhang et al., 2009*). Therefore, *CDON* mutations likely require additional insults to contribute to HPE. Studies with mice are consistent with this conclusion. *Cdon*-/- mice on a 129S6 genetic background have a subthreshold defect in HH signaling and are subject to HPE induced by genetic and environmental modifiers; among the latter is ethanol (EtOH) (*Hong and Krauss, 2018*). *Cdon* mutation or in utero EtOH exposure individually yielded little effect on 129S6 mice. The combination, however, synergized to inhibit HH signaling in the developing forebrain and produced a complete spectrum of HPE phenotypes with high penetrance (*Hong and Krauss, 2012*). Furthermore, fetal EtOH induced a low penetrance of HPE in C57BL/6J mice, and this was exacerbated by heterozygosity for *Shh* or *Gli2* (*Kietzman et al., 2014*). Consistent with the notion that a threshold of HH signaling activity is rate-limiting in midline patterning, genetic removal of one copy of the negative pathway regulator, *Ptch1*, rescued 129S6 *Cdon*-/- mice from EtOH-induced HPE (*Hong and Krauss, 2013*).

Defective HH pathway function may be a final common mechanism for all classical forms of HPE, so these results do not prove that HH signaling is the direct synergistic target of *Cdon* mutation and in utero EtOH exposure. Several lines of evidence argue that EtOH itself, rather than a consequence of its metabolism, is the HPE-inducing teratogen (*Hong and Krauss, 2017*). In this study, we demonstrate that the window of sensitivity to EtOH-induced HPE in 129S6 *Cdon*-/- mice is closed by embryonic day (E) 7.5, with the sensitivity period overlapping Nodal-mediated specification of the PCP from the APS. Furthermore, we find that CDON interacts genetically and physically with regulators of Nodal signaling. Finally, EtOH inhibits Activin/Nodal pathway signaling in vitro in mouse epiblast stem cells (mEpiSCs), which have transcriptional and functional properties that resemble APS cells (*Kojima et al., 2014*; *Tsakiridis et al., 2014*). Together these results argue that, unexpectedly, *Cdon* mutation and fetal alcohol synergize to induce HPE by interfering with Nodal signaling. These results illuminate mechanisms of gene-environment interaction in a high fidelity, multifactorial model of a common birth defect.

## Results

### The window of sensitivity to EtOH-induced HPE coincides with Nodal signaling

All studies were performed with mice on a 129S6 background and are referred to only by genotype unless otherwise noted. Our model uses one-hour timed mating of *Cdon*+/- mice, allowing rigorous identification of a window of sensitivity for EtOH's teratogenic effects. The standard EtOH treatment regimen involves one dose at E7.0 and a second dose four hours later. This results in a full range of HPE phenotypes in *Cdon*-/- mice with overall penetrance of ~75% (*Hong and Krauss, 2012*). The protocol is analogous to the original regimen developed to model fetal alcohol spectrum disorders in C57BL/6J mice, wherein ~ 19% developed HPE phenotypes (*Aoto et al., 2008*; *Sulik et al.,*

*1981*). HPE phenotypes were not induced in wild type C57BL/6J mice when EtOH was administered at E6.5 or E7.5 (*Aoto et al., 2008*). We have shown that EtOH is ineffective in *Cdon*-/- mice when administered at E8.0 (*Hong and Krauss, 2012*). To further refine the temporal window of sensitivity, we assessed the ability of EtOH to induce HPE in *Cdon*-/- animals when administered at E7.25 and E7.5, with embryos examined for external HPE phenotypes at E14.0. EtOH was effective at E7.25, but not at E7.5 (*Table 1* and *Figure 1*).

EtOH itself is the HPE-inducing teratogen in this system (*Hong and Krauss, 2017*). The developmental processes directly perturbed by EtOH must therefore occur: 1) during the sensitive time window; and 2) prior to EtOH clearance, even if defects in midline development occur later. When administered at E7.0, maternal blood EtOH levels peaked at E7.25 and were no longer detectable by ~E7.75 (*Hong and Krauss, 2012*). Critically, EtOH is no longer effective at E7.5, a time reported to be prior to SHH expression and function in prechordal plate-mediated rostroventral midline patterning (*Echelard et al., 1993*; *McMahon et al., 2003*). These results argue that EtOH targets processes other than, or in addition to, HH signaling in *Cdon*-/- mice. Consistent with this conclusion, the time of peak sensitivity to HPE induced by the direct SMO inhibitor, Vismodegib, was E7.5, 6–12 hr later than for EtOH (*Heyne et al., 2015*).

Both the window of sensitivity to EtOH and peak EtOH levels overlap with induction of the PCP by Nodal signaling. We previously observed that expression of *Foxa2* and *Gsc*, two Nodal target genes that mark the APS (from which the PCP is derived), was diminished at E7.25 in EtOH-treated *Cdon*-/- embryos (*Hong and Krauss, 2012*). Both *Cdon* mutation and EtOH exposure were required for this effect. Expression of *Lefty2*, a direct Nodal pathway target gene, was also reduced at E7.25 specifically in EtOH-treated *Cdon*-/- embryos (*Figure 2*). Taken together, these results showed that the synergy between *Cdon* mutation and in utero EtOH exposure occurred during Nodal-dependent specification of the PCP and that reduction of Nodal target gene expression required a combination of mutation and teratogen. The observation that loss of CDON was required for the effect suggests that, in addition to its role as a SHH co-receptor, CDON functions earlier in development to promote midline patterning, potentially via the Nodal pathway. In fact, *Cdon* expression initiates during gastrulation and is seen in embryonic mesoderm and ectoderm (*Hong and Krauss, 2012*; *Mulieri et al., 2000*).

## *Cdon* interacts genetically with Nodal pathway components

Mice with mutations resulting in complete loss of Nodal signaling gastrulate abnormally and die early in development, but hypomorphic pathway mutants display a range of HPE phenotypes (*Andersson et al., 2006*; *Chu et al., 2005*; *Lowe et al., 2001*; *Nomura and Li, 1998*; *Schier, 2009*; *Shen, 2007*; *Song et al., 1999*). To further probe the linkage between CDON and regulation of Nodal signaling, we assessed whether *Cdon* interacted genetically with *Tdgf1* and *Lefty2*, direct positive and negative regulators of the Nodal pathway, respectively. When analyzed at E10,~14% of *Cdon*-/- embryos treated with EtOH had alobar HPE and cyclopia, dying in utero by E11 (*Hong and Krauss, 2012*; *Hong and Krauss, 2013*). When studied at E14,~70% of such embryos displayed lobar HPE and a range of craniofacial midline phenotypes, including fused upper lip and single nostril (*Hong and Krauss, 2012*; *Hong and Krauss, 2013*). To address genetic interactions between *Cdon* and Nodal pathway regulators, we took advantage of the high penetrance of phenotypes at E14 and analyzed embryos at this stage for these unambiguous, easily scored phenotypes.

**Table 1.** Time course of EtOH-Induced HPE in *Cdon*-/- Embryos

| EtOH treatment: | E7.25 | | | | E7.5 | |
| --- | --- | --- | --- | --- | --- | --- |
| | Saline | | EtOH | | EtOH | |
| Phenotype* | *Cdon*+/- | *Cdon*-/- | *Cdon*+/- | *Cdon*-/- | *Cdon*+/- | *Cdon*-/- |
| Fused upper lip | 0/17 | 2/14 | 0/21 | 13/24** | 0/13 | 0/22 |
| Single nostril | 0/17 | 0/14 | 0/21 | 5/24 | 0/13 | 0/22 |
| Proboscis | 0/17 | 0/14 | 0/21 | 2/24 | 0/13 | 0/22 |

* All embryos with HPE had fused upper lip, a fraction of these showed single nostril and proboscis.
**p=0.0165 by Fisher's two-tailed exact test, when compared to EtOH-treated *Cdon*+/- embryos.

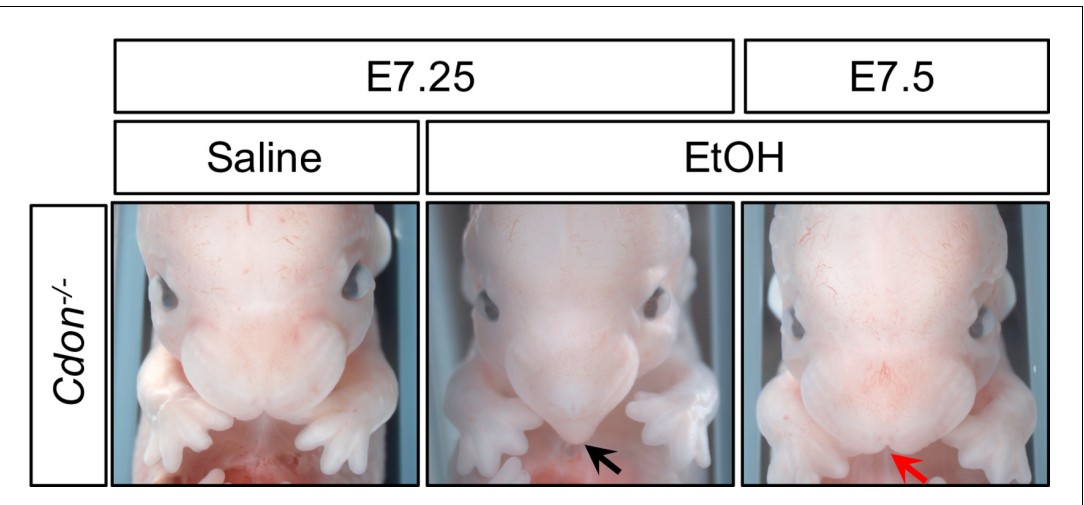

**Figure 1.** Time course of EtOH-Induced HPE in *Cdon*[-/-] Embryos. Frontal views of E14.0 embryos. Treatment of *Cdon*[-/-] embryos with EtOH at E7.25, but not E7.5, results in HPE (see *Table 1* for quantification). The *Cdon*[-/-] embryo treated with EtOH at E7.25 displays a fused upper lip and single nostril (black arrow), whereas the *Cdon*[-/-] embryo treated with EtOH at E7.5 does not and resembles the saline control (red arrow).

Targeted mutations in *Tdgf1* and *Lefty2* (*Ding et al., 1998*; *Meno et al., 1999*) were placed onto the 129S6 genetic background and then crossed to *Cdon* mutants to generate double heterozygotes. Offspring of intercrosses of these mice were further crossed and pregnant females were treated with EtOH or saline as a control. *Tdgf1* is essential for Nodal signaling. *Lefty2* is induced in response to Nodal signaling to provide critical negative feedback as an inhibitor of the pathway. Since CDON and EtOH may act together at the level of Nodal signaling in HPE, we predicted that removal of one copy of *Tdgf1* would sensitize *Cdon*[-/-] embryos to EtOH-induced HPE. In contrast, removal of one copy of *Lefty2* would be predicted to rescue *Cdon*[-/-] embryos from EtOH-induced HPE. (Null mutations in *Tdgf1* and *Lefty2* result in early lethality due to strong gastrulation defects (*Ding et al., 1998*; *Meno et al., 1999*), so studying homozygous double mutants with *Cdon* is impossible.) To permit the detection of sensitization by *Tdgf1* heterozygosity, we used a dose of EtOH which we previously reported gives ~30% penetrance of HPE at E14 (2.9 g/kg) (*Hong and*

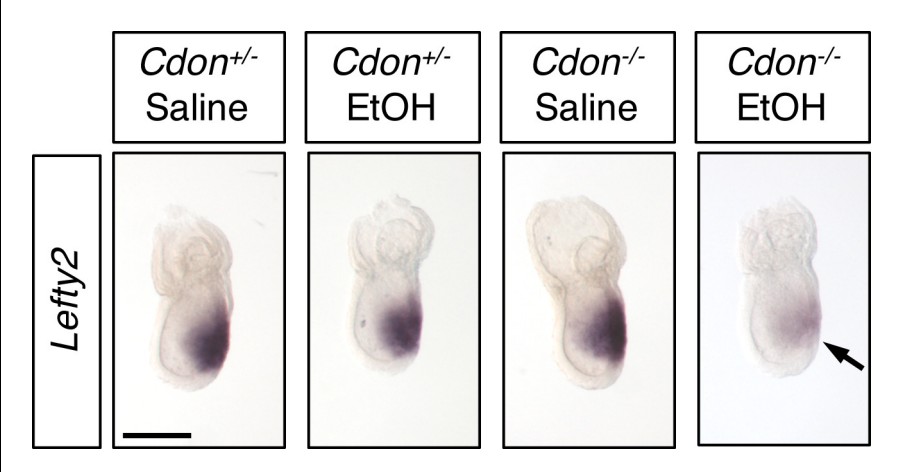

**Figure 2.** Defective Expression of *Lefty2* at the Primitive Streak Stage of EtOH-Treated *Cdon*[-/-] Embryos. Whole mount in situ hybridization analyses of *Lefty2* in embryos of the indicated genotype and treatment harvested at E7.25. *Lefty2* expression was decreased specifically in EtOH-treated *Cdon*[-/-] embryos. Numbers of embryos with similar results: *Cdon*[+/-] (saline) = 6; *Cdon*[+/-] (EtOH) = 4; *Cdon*[-/-] (saline) = 9; *Cdon*[-/-] (EtOH) = 9. Scale Bar, 200 μm.

*Krauss, 2012*). With this regimen, 28% of EtOH-treated $Cdon^{-/-};Tdgf1^{+/+}$ embryos had HPE, whereas 55% of $Cdon^{-/-};Tdgf1^{+/-}$ embryos did (p<0.014) (*Table 2* and *Figure 3*). With the exception of a single EtOH-treated $Cdon^{+/-};Tdgf1^{+/-}$ embryo, EtOH-treated mice of other genotypes and saline-treated control mice did not display HPE. Removal of one copy of *Tdgf1* therefore sensitized $Cdon^{-/-}$ mice to EtOH-induced HPE.

$Nodal^{-/-}$ and $Lefty2^{-/-}$ embryos have opposite defects in gastrulation and display dosage-dependent genetic interactions, with *Lefty2* functioning as a negative regulator of the pathway (*Meno et al., 1999*). To test for phenotypic suppression in EtOH-induced HPE, we used the standard dose of EtOH (3.48 g/kg) for these analyses, as it results in high penetrance. All embryos with HPE at E14 display a fused upper lip, while only some show the more severe single nostril phenotype. Approximately 70% of EtOH-treated $Cdon^{-/-};Lefty2^{+/+}$ embryos had a fused upper lip, similar to our previous findings (*Hong and Krauss, 2012*; *Hong and Krauss, 2013*), whereas 54% of $Cdon^{-/-};Lefty2^{+/-}$ embryos did, revealing a trend toward overall reduction of HPE frequency (p=0.14). However, while 21% of EtOH-treated $Cdon^{-/-};Lefty2^{+/+}$ embryos had a single nostril, only 6% of $Cdon^{-/-};Lefty2^{+/-}$ embryos did (p=0.04) (*Table 3* and *Figure 4*). Again, EtOH-treated mice of other genotypes and saline-treated control mice did not display HPE. Therefore, removal of one copy of *Lefty2* rescued a more severe form of HPE associated with EtOH treatment. Taken together, these studies revealed that the Nodal signaling components *Tdgf1* and *Lefty2* act as heterozygous enhancer and suppressor genes, respectively, of EtOH-induced HPE in $Cdon^{-/-}$ embryos.

## EtOH inhibits Activin/Nodal signaling in mEpiSCs

The APS is a small, transiently existing cell population, making it hard to study directly. Cultured mEpiSCs have transcriptional and functional properties resembling APS cells (*Kojima et al., 2014*; *Tsakiridis et al., 2014*), making them a tractable in vitro surrogate model. mEpiSCs rely on FGF2 and Activin/Nodal signaling for maintenance as self-renewing, pluripotent stem cells (*Brons et al., 2007*; *Vallier et al., 2009*). Activin A and Nodal use the same receptor and signaling mechanism, except that Nodal, but not Activin A, requires the co-receptor TDGF1. Therefore, we evaluated the effects of EtOH on Activin/Nodal signaling in mEpiSCs.

A direct readout of Nodal signaling activity is receptor-mediated phosphorylation of SMAD2 at its C-terminus (*Robertson, 2014*; *Schier, 2009*; *Shen, 2007*). We assessed C-terminally phosphorylated SMAD2 (p-SMAD2C) and total SMAD2 levels in the mEpiSC line, EpiSC9 (*Najm et al., 2011*), after treatment with EtOH for 6 hr. A relatively short treatment duration was chosen because: 1) the in vivo model is an acute exposure regimen; 2) the window of sensitivity to EtOH-induced HPE is <12 hr; and 3) high EtOH levels last only about 10 hr in pregnant females (*Hong and Krauss, 2012*). EtOH dose-dependently diminished p-SMAD2C levels in mEpiSCs, without altering total SMAD2 levels (*Figure 5A,B*).

EtOH stimulates activation of the MAP kinase, JNK, in specific cell types (*McAlhany et al., 2000*; *Ren et al., 2017*). SMAD2 is phosphorylated in its linker region by MAP kinases (*Massague, 2003*; *Rezaei et al., 2012*), usually leading to inhibition of SMAD2 function (*Grimm and Gurdon, 2002*;

**Table 2.** *Tdgf1* Heterozygosity Enhances EtOH-Induced HPE in $Cdon^{-/-}$ Embryos.

| Treatment | Genotype (# embryos with HPE/total (%))* | | |
|---|---|---|---|
| | $Cdon^{+/-};Tdgf1^{+/-}$ | $Cdon^{-/-};Tdgf1^{+/+}$ | $Cdon^{-/-};Tdgf1^{+/-}$ |
| Saline | 0/23 (0%) | 0/11 (0%) | 2/27 (7.4%) |
| EtOH (2.9 g/kg) | 1/48 (2.1%) | 9/32 (28.1%) | 31/56 (55.4%)** |

* Crosses between the following genotypes were used to generate the genotypes scored above:

$Cdon^{+/-};Tdgf1^{+/-}$ x $Cdon^{+/-};Tdgf1^{+/-}$.

$Cdon^{+/-};Tdgf1^{+/-}$ x $Cdo-^{+/-};Tdgf1^{+/-}$.

$Cdon^{+/-}$ x $Cdon^{+/-};Tdgf1^{+/-}$.

$Cdon^{+/-}$ x $Cdon^{-/-};Tdgf1^{+/-}$.

$Cdon^{-/-}$ x $Cdon^{+/-};Tdgf1^{+/-}$.

No HPE was found in offspring genotypes other than those shown.

** p=0.014 by Fisher's two-tailed exact test, when compared to EtOH-treated $Cdon^{-/-}; Tdgf1^{+/+}$ embryos.

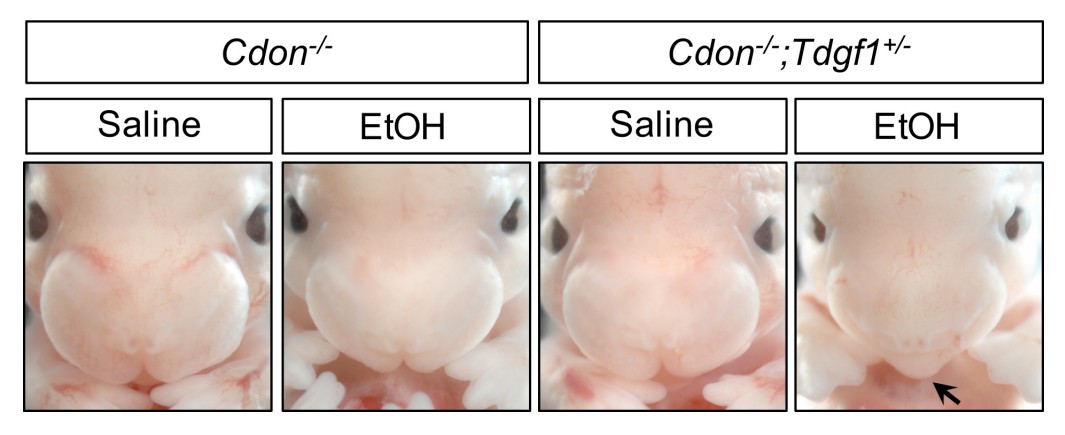

**Figure 3.** *Tdgf1* Heterozygosity Enhances EtOH-Induced HPE in *Cdon⁻/⁻* Embryos. Frontal views of E14.0 embryos. Removal of one copy of *Tdgf1* enhanced the response of *Cdon⁻/⁻* embryos to a dose of 2.9 g/kg EtOH (see **Table 2** for quantification). The EtOH-treated *Cdon⁻/⁻;Tdgf1⁺/⁻* embryo displays a fused upper lip (arrow).

*Kretzschmar et al., 1999*; *Lessard et al., 2018*). We therefore assessed total and phosphorylated (activated) forms of JNK1 MAP kinase in EtOH-treated mEpiSCs. EtOH dose-dependently increased phospho-JNK1 (T183/Y185) levels (p-JNK1; *Figure 5A,B*). Consistent with EtOH-induced p-JNK1 activation, phosphorylation of the SMAD2 linker region (p-SMAD2L) also occurred in an EtOH dose-dependent manner, with production of p-JNK1 and p-SMAD2L correlating well (*Figure 5A,B*).

Human embryonic stem cells (hESCs) resemble mEpiSCs (*Pauklin and Vallier, 2015*). Treatment of hESC or mEpiSC cultures with the ALK4 inhibitor, SB43152, led to reduction of Activin/Nodal target gene expression and, eventually, induction of early markers of neuroectoderm differentiation (*Chng et al., 2010*; *Li et al., 2015*; *Liu et al., 2018*; *Vallier et al., 2009*). These studies generally used time courses of days to weeks, but Vallier et al. assessed expression of five Activin/Nodal target and/or pluripotency-related genes in hESCs treated with SB43152 for 6 hr, the same amount of time we treated mEpiSCs with EtOH. Levels of *Nanog*, *Nodal*, and *LeftyA* mRNA were reduced by SB43152, whereas levels of *Pou5f1* (also called *Oct4*) and *Sox2* underwent little or no change (*Vallier et al., 2009*). We assessed expression of these five genes in mEpiSCs treated with EtOH for 6 hr (*Figure 5C*). Similar to SB43152-treated hESCs, EtOH-treated mEpiSCs displayed significantly reduced levels of *Nanog* and *Nodal* mRNAs, while *Pou5f1* and *Sox2* mRNA levels were largely unchanged. Expression of the additional SB43152-inhibitable mEpiSC markers *Fgf5* and *T* (*Liu et al., 2018*) were also reduced by EtOH treatment (*Figure 5C*). In contrast, EtOH did not alter levels of

**Table 3.** *Lefty2* Heterozygosity Supresses EtOH-Induced HPE in *Cdon⁻/⁻* Embryos.

| Treatment | Genotype (# affected/total (%))* | | | |
|---|---|---|---|---|
| | Total HPE | | Single nostril | |
| | *Cdon⁻/⁻;Lefty2⁺/⁺* | *Cdon⁻/⁻;Lefty2⁺/⁻* | *Cdon⁻/⁻;Lefty2⁺/⁺* | *Cdon⁻/⁻;Lefty2⁺/⁻* |
| Saline | 1/12 (8.3%) | 0/10 (0%) | 0/12 (0%) | 0/10 (0%) |
| EtOH (3.48 g/kg) | 33/47 (70.2%) | 26/48 (54.2%) | 10/47 (21.3%) | 3/48 (6.3%)** |

* Crosses between the following genotypes were used to generate the genotypes scored above:

*Cdon⁺/⁻;Lefty2⁺/⁻* x *Cdon⁺/⁻;Lefty2⁺/⁻*.

*Cdon⁺/⁻;Lefty2⁺/⁻* x *Cdo-⁺/⁻;Lefty2⁺/⁻*.

*Cdon⁺/⁻* x *Cdon⁺/⁻;Lefty2⁺/⁻*.

*Cdon⁺/⁻* x *Cdo-⁻/⁻;Lefty2⁺/⁻*.

*Cdon⁻/⁻* x *Cdon⁺/⁻;Lefty2⁺/⁻*.

No HPE was found in offspring genotypes other than those shown.

**p=0.04 by Fisher's two-tailed exact test, when compared to EtOH-treated *Cdon⁻/⁻;Lefty2⁺/⁺* embryos with a single nostril.

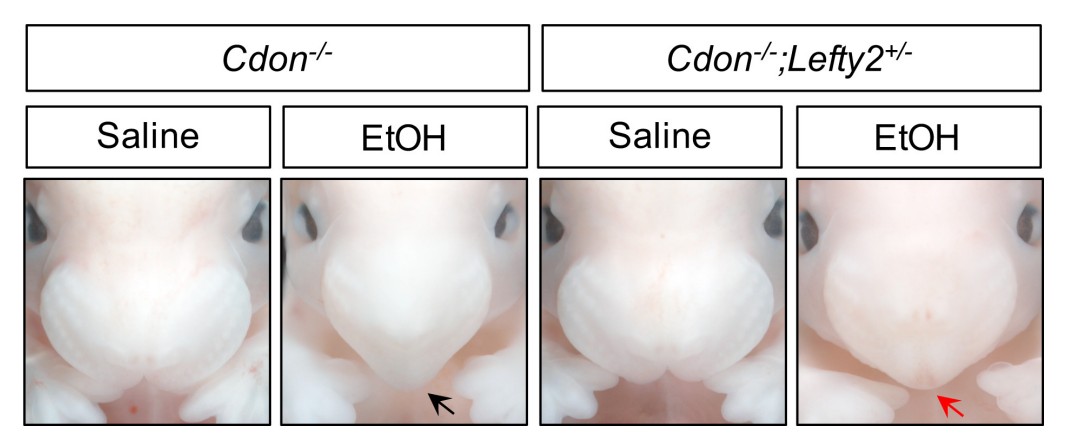

**Figure 4.** *Lefty2* Heterozygosity Suppresses EtOH-Induced HPE in *Cdon^-/-^* Embryos. Frontal views of E14.0 embryos. Removal of one copy of *Lefty2* suppressed formation of single nostril in *Cdon^-/-^* embryos in response to a dose of 3.48 g/kg EtOH (see *Table 3* for quantification). The EtOH-treated *Cdon^-/-^* embryo displays a fused upper lip and single nostril (black arrow), whereas the EtOH-treated *Cdon^-/-^;Lefty2^+/-^* embryo has a fused upper lip and two nostrils (red arrow).

*Lefty1* or *Lefty2* in mEpiSC cultures (*Figure 5—figure supplement 1*). As these are known Activin/Nodal target genes, this result might be related to differences in SB43152 vs. EtOH treatment, or differences in half-lives of these mRNAs in human vs. mouse cultures. Therefore, acute EtOH treatment of mEpiSCs resulted in an inhibitory SMAD2 phosphorylation pattern and changes in gene expression similar to those induced by a direct Activin/Nodal pathway inhibitor. Induction of neuroectoderm-specific genes (e.g., *Hoxa1*, *Six1*, and *Gbx2*) was observed in hESCs treated with SB43152 for 2–6 days (*Chng et al., 2010*; *Vallier et al., 2009*). After 6 hr of EtOH treatment, we found a varied response, with *Hoxa1* levels strongly induced, *Six1* expression displaying a biphasic dose-response, and *Gbx2* levels reduced (*Figure 5—figure supplement 1*).

### *Cdon* and *Lrp2* genetically interact in mice to generate phenotypes resembling Nodal pathway hypomorphs, and both bind to TDGF1

Our findings argue that CDON plays an earlier role in development than its known function as a SHH co-receptor. The need for EtOH exposure to reveal this role suggests that CDON's effects in primitive streak embryos may be redundant with other factors. The HH co-receptors BOC and GAS1 are not likely to be such factors, as *Cdon;Boc;Gas1* triple knockout mice had a phenotype that is close to a complete loss of embryonic HH pathway function, but not Nodal pathway-like phenotypes (*Allen et al., 2011*). LRP2 is an endocytic and auxiliary receptor for multiple morphogenetic ligands and receptors (*Willnow and Christ, 2017*). *Lrp2^-/-^* mice have HPE of variable penetrance and severity (*Christ et al., 2012*; *Spoelgen et al., 2005*; *Willnow and Christ, 2017*; *Willnow et al., 2012*). We therefore constructed mice with mutations in *Cdon* and *Lrp2*. Some *Cdon;Lrp2* double mutants at E11.5 had a severe truncation of anterior head and face structures, a phenotype neither single mutant displayed (*Figure 6A*). Double-mutant embryos studied at several stages displayed a range of phenotypes, including milder craniofacial truncation and strong HPE (*Figure 6B,C*). Of 30 *Cdon; Lrp2* double mutants scored between 20 and 37 somites, 24 (80%) had HPE and 6 (20%) displayed anterior truncations. Loss of anterior head structure is a more severe phenotype than that seen even in *Smo* mutants, which lack all HH signaling (*Zhang et al., 2001*). These phenotypes – anterior truncations and HPE – are similar to those observed in mice with partial loss of Nodal pathway function (e.g., hypomorphic *Nodal* and *Tdgf1* mutants, and *Nodal^+/-^;Smad2^+/-^* mutants, each have phenotypes that include truncation of anterior head structures and severe HPE) (*Andersson et al., 2006*; *Chu et al., 2005*; *Lowe et al., 2001*; *Nomura and Li, 1998*; *Song et al., 1999*). The *Cdon/Lrp2* genetic interaction is specific and selective: the *Cdon* paralog *Boc* is not expressed in primitive streak-stage embryos (*Mulieri et al., 2002*; *Zhang et al., 2001*), and *Boc* mutant mice do not synergize with *Lrp2* mutants (unpublished results), nor are they sensitive to EtOH-induced HPE (*Hong and Krauss, 2012*).

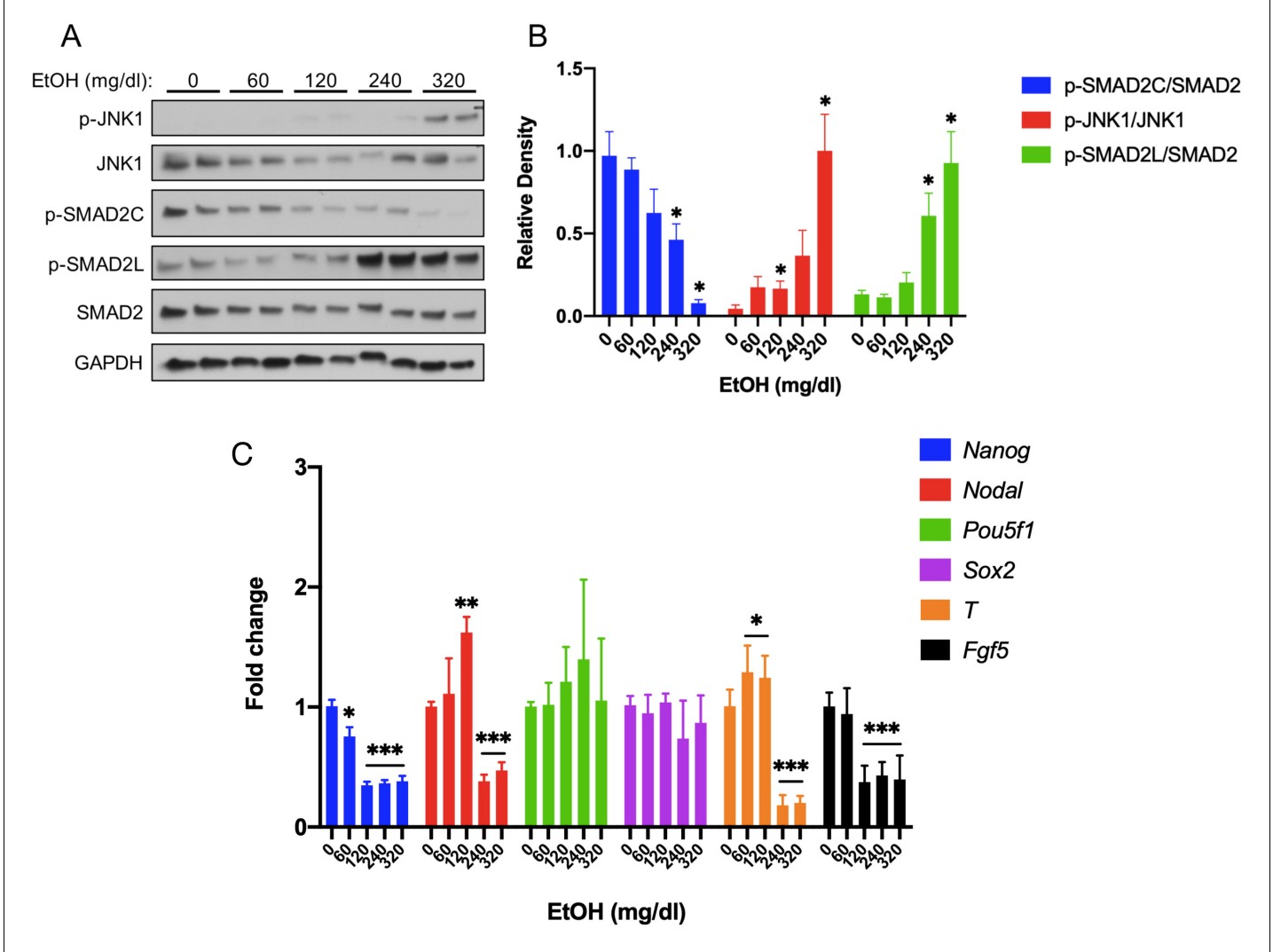

**Figure 5.** Acute EtOH treatment inhibits activin/nodal signaling in mEpiSCs. (**A**) Representative western blot of mEpiSCs treated with the indicated doses of EtOH for 6 hr. GAPDH was used as a loading control. (**B**) Densitometric quantification of p-SMAD2C levels, relative to total Smad2; p-JNK1 levels, relative to total JNK1; and p-SMAD2L levels, relative to total SMAD2, with the indicated doses of EtOH. N = 3 experiments with two biological replicates (as shown in (**A**)) in each experiment. (**C**) qRT-PCR analysis of *Nanog*, *Nodal*, *Pou5f1*, and *Sox2* expression in mEpiSCs treated with the indicated doses of EtOH for 6 hr. Expression was normalized to *Gapdh* expression. N = 3 experiments with two biological replicates in each experiment. Values for (**B**) and (**C**) are means ± SEM, *p<0.05, **p<0.01, ***p,0.001 by Student's t-test.

The online version of this article includes the following source data and figure supplement(s) for figure 5:

**Source data 1.** Source data for quantification of western blot results shown in *Figure 5A and B*.

**Source data 2.** Source data for quantification of qRT-PCR results shown in Figure 5C.

**Figure supplement 1.** Analysis of gene expression in EtOH-treated mEpiSCs.

**Figure supplement 1—source data 1.** Source data for quantification of qRT-PCR results shown in *Figure 5*.

CDON is a multifunctional co-receptor and promotes HH-independent signaling when associated with various other receptors and cell adhesion molecules. In addition to binding HH ligands and PTCH1, CDON interacts with and influences signaling by classical cadherins; the Netrin receptor, Neogenin; and the WNT co-receptor, LRP6 (*Bae et al., 2011*; *Bae et al., 2009*; *Izzi et al., 2011*; *Jeong et al., 2014*; *Kang et al., 2003*; *Kang et al., 2004*; *Lu and Krauss, 2010*; *Tenzen et al., 2006*). We hypothesized that CDON might work similarly in Nodal signaling. To address this possibility, we assessed whether CDON binds to Nodal receptor components. A secreted CDON ectodomain fused in-frame at its carboxy terminus with the IgG Fc region (CDON-Fc) was expressed in

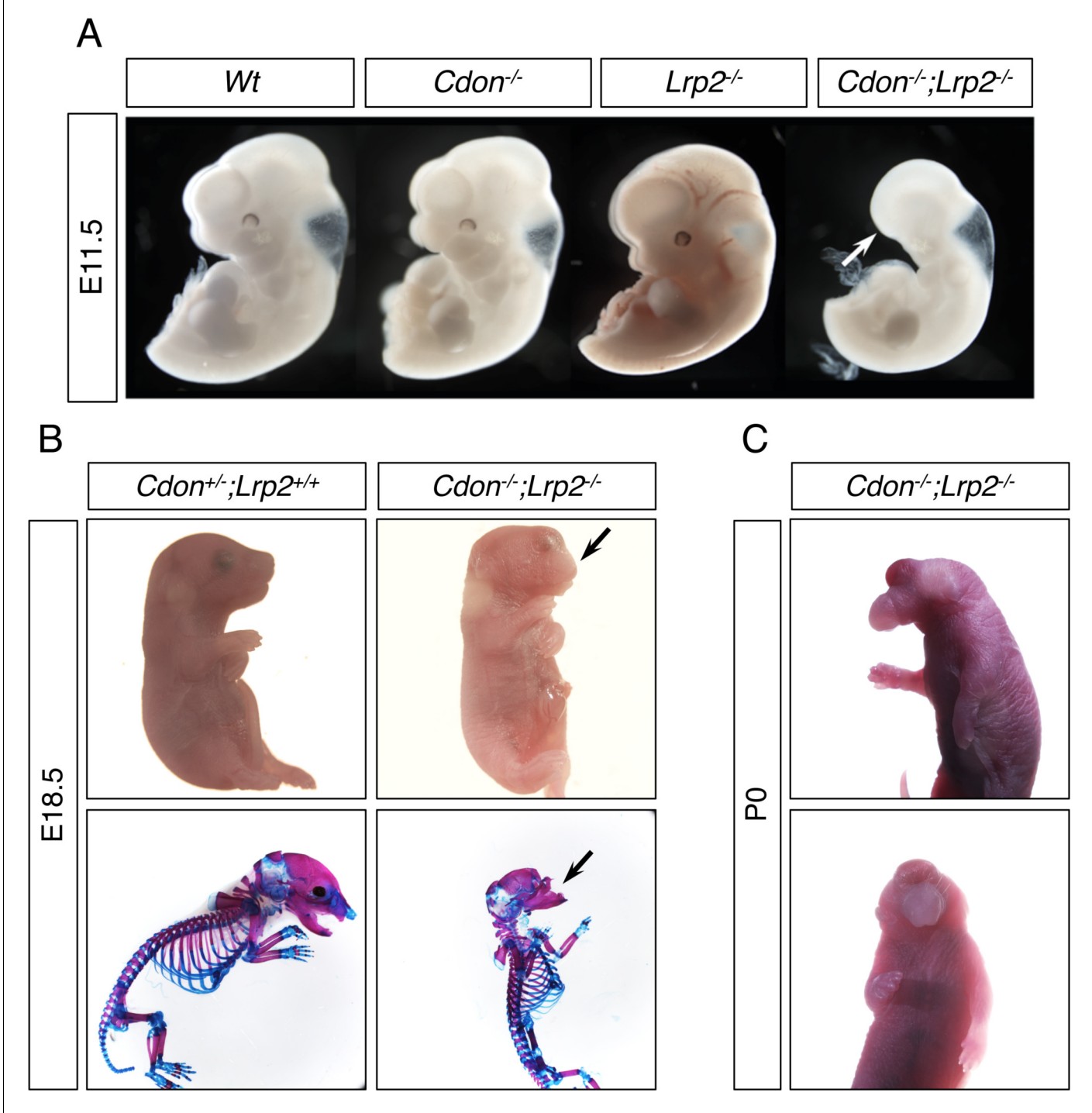

**Figure 6.** *Cdon* and *Lrp2* Interact Genetically to Produce HPE and Anterior Truncations in Mice. (**A**) Whole mount E11.5 embryos of the indicated genotype. Note the loss of anterior head structures in the *Cdon⁻/⁻;Lrp2⁻/⁻* embryo (arrow). (**B**) E18.5 embryos of the indicated genotype in whole mount and alizarin red/alcian blue-stained skeleton preparations. Note the craniofacial truncation in the *Cdon⁻/⁻;Lrp2⁻/⁻* embryos (arrows). (**C**) Whole mount P0 *Cdon⁻/⁻;Lrp2⁻/⁻* embryo showing severe HPE.

HEK293T cells with individual ectodomains of the Nodal receptor (ActRIIA, ActRIIB, ALK4, and TDGF1) fused in-frame at their carboxy termini with alkaline phosphatase (AP). As positive and negative controls, SHH-N-AP and CD164-AP were used, respectively (SHH-N is the active portion of SHH, CD164 is a cell surface sialomucin) (*Kang et al., 2002*; *Tenzen et al., 2006*). The use of secreted

ectodomains reduces the possibility that unknown, cell surface-bound factors promote binding or contribute directly to the complex. Conditioned medium was collected, and equivalent amounts of AP fusion proteins from conditioned media were pulled down and blotted for captured CDON-Fc. Conversely, equivalent levels of Fc were pulled down, and associated AP activity was quantified. Similar results were obtained in each pull-down: CDON-Fc interacted efficiently with TDGF1-AP, similar to the interaction between CDON-Fc and SHH-N-AP (*Figure 7A,B*). CDON-Fc did not bind to the negative control CD164-AP, and it interacted only weakly with the AP-tagged versions of ALK4, ActRIIA, and ActRIIB (*Figure 7A,B*).

As CDON and LRP2 interacted genetically to produce Nodal hypomorph-like phenotypes, we asked whether LRP2, like CDON, binds to TDGF1. The extracellular region of LRP2 harbors four repetitive modules, each comprising a cluster of complement-type repeats followed by EGF-type repeats and β-propellers. Each module is likely capable of independently binding ligands, as shown for the related receptor, LRP1 (*Croy et al., 2003*). LRP2 is a huge polypeptide of 4660 amino acids, making it difficult to express in vitro. Therefore, we constructed a series of soluble 'mini-receptors' (designated sR1, sR2, sR3, and sR4) spanning virtually the entire ectodomain in four non-overlapping pieces, each fused in-frame at their carboxy termini to the IgG Fc region (see *Figure 7C*). Each soluble ectodomain segment and TDGF1-AP were expressed in HEK293T cells and assessed for their ability to bind to each other. LRP2 binds SHH-N (*Christ et al., 2012*; *McCarthy et al., 2002*), so this ligand was used again as a positive control, with CD164-AP serving as a presumptive negative control. Each soluble LRP2 mini-receptor ectodomain (i.e., sR1 to sR4) pulled down SHH-N-AP, revealing that LRP2 harbors multiple SHH-N binding sites (*Figure 7D*). These results were confirmed by expressing each ectodomain segment joined with the native LRP2 transmembrane and cytoplasmic domains (mini-receptors R1, R2, R3, and R4; *Figure 7C*) in NIH3T3 cells. These cultures were incubated with recombinant SHH-N-GST fusion proteins for 2 hr, followed by immunoprecipitation with anti-LRP2 antibody recognizing all mini-receptors and western blotting with antibodies to LRP2 or GST. Again, all four mini-receptors pulled down SHH-N, whereas no pulldown of SHH-N was seen in control transfectants lacking LRP2 mini-receptors (*Figure 7—figure supplement 1*). Finally, we assessed LRP2 binding to TDGF1 with the soluble ectodomain binding assay. Similar to its multivalent interaction with SHH-N, the soluble LRP2 ectodomain segments sR3 and sR4 pulled down TDGF1-AP (*Figure 7D*). Soluble LRP2 ectodomain segments sR1 and sR2 also pulled down TDGF1-AP, but greater variability in the interaction was observed, leading to $p > 0.05$ ($p = 0.14$ and $p = 0.06$, respectively). The efficiency of LRP2 mini-receptor ectodomains to bind SHH-N-AP and TDGF1-AP was similar to that of CDON-Fc (*Figure 7D*).

## Discussion

HPE is a common developmental disorder, but its etiology remains poorly understood. Mutations in the Nodal, HH, and FGF pathways are associated with HPE (*NISC Comparative Sequencing Program et al., 2018*), but these alone are insufficient to drive aberrant development. Identifying and understanding the critical phenotypic modifiers of such mutations is important but still in its infancy. Many birth defects are thought to be caused by poorly defined interactions between genetic and environmental risk factors, but the mechanistic bases of these interactions are largely unknown (*Krauss and Hong, 2016*). *Cdon* mutation and fetal alcohol are each implicated in human HPE (*Abe et al., 2018*; *Bae et al., 2011*; *Cohen and Shiota, 2002*; *Croen et al., 2000*; *NISC Comparative Sequencing Program et al., 2018*), though neither may be sufficient. The *Cdon* mutation plus fetal alcohol model is noteworthy for its specificity and fidelity to many aspects of human HPE (*Hong and Krauss, 2012*). Therefore, illumination of how loss of *Cdon* interacts with in utero EtOH exposure provides insight into mechanisms of HPE and how fetal alcohol functions as a teratogen.

The Nodal pathway is developmentally upstream from the HH pathway in patterning the rostro-ventral midline. While CDON is clearly a component of the HH pathway, it regulates signaling within several additional pathways as well (*Bae et al., 2011*; *Bae et al., 2009*; *Izzi et al., 2011*; *Jeong et al., 2014*; *Kang et al., 2003*; *Kang et al., 2004*; *Lu and Krauss, 2010*; *Tenzen et al., 2006*). Multiple lines of evidence presented here argue that CDON also regulates Nodal signaling and that the Nodal pathway is a major point of synergistic interaction between mutation of *Cdon* and fetal EtOH exposure. The evidence includes: (1) EtOH-treated *Cdon*⁻/⁻ embryos display defects

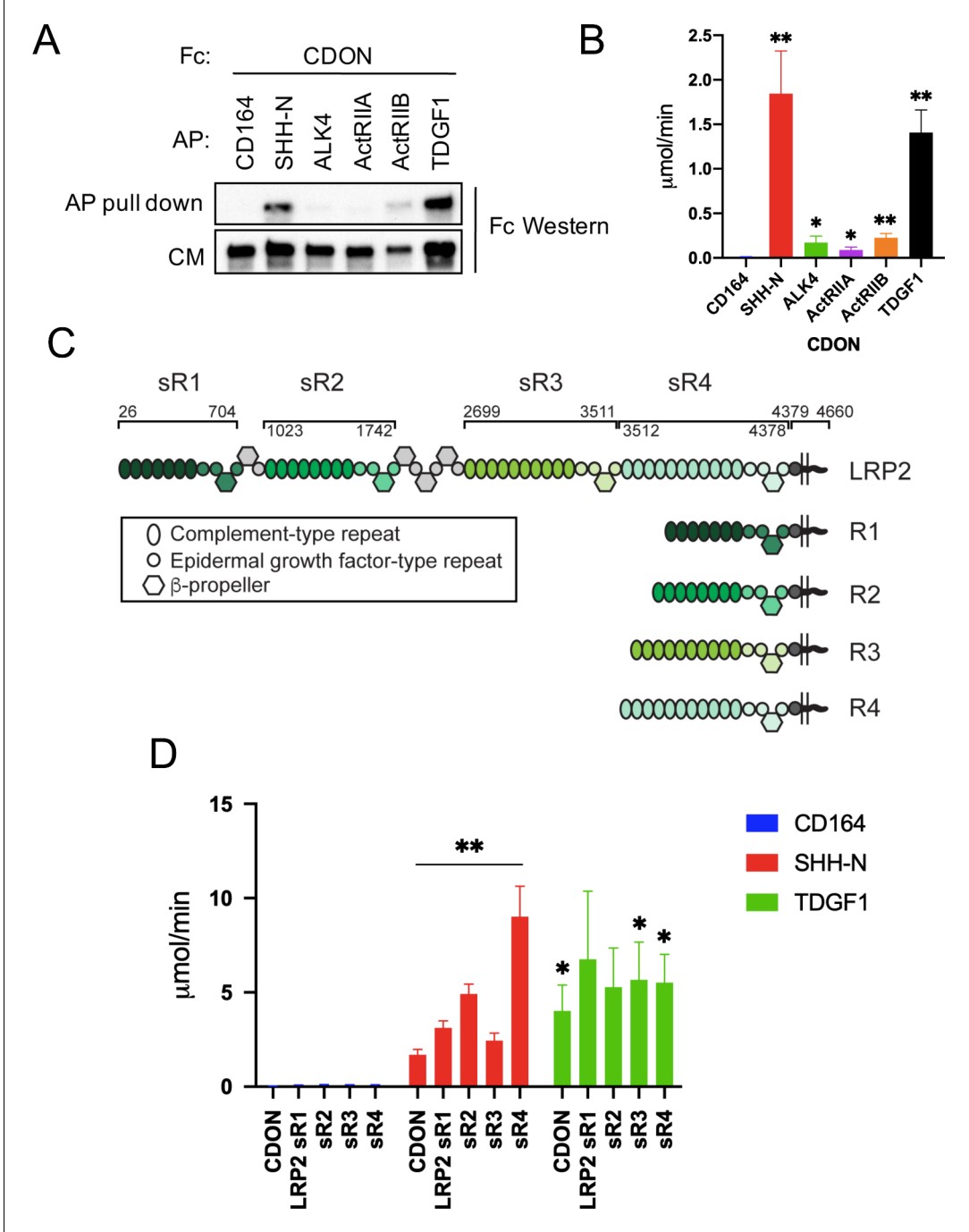

**Figure 7.** CDON and LRP2 Bind to TDGF1. (**A**) The indicated AP-tagged proteins were co-expressed with CDON-Fc, and equivalent amounts of AP proteins in conditioned medium (CM) were pulled down. Levels of CDON-Fc in CM and in the pull down were detected by western blot. (**B**) CDON-Fc was pulled down from CM derived from transfections as in (**A**), the amount of CDON-Fc from various CM normalized, and associated AP enzyme activity quantified, N = ≥4 for (**A**) and (**B**). (**C**) Schematic of full length LRP2 and recombinant mini-receptor variants used. Each mini-receptor spans one of the four repetitive modules of the extracellular receptor domain and was expressed either as soluble ectodomain fragments fused to IgG Fc (designated sR1- sR4) or as a membrane-bound mini-receptor fused to the authentic transmembrane and intracellular domains of LRP2 (designated R1-R4). Soluble ectodomain fragments sR1 – sR4 were studied in panel (**D**), whereas mini-receptors R1 – R4 were used in *Figure 7—figure supplement 1* . Amino acid numbers next to the full-length receptor polypeptide indicate the extent of individual receptor domains. (**D**) The indicated Fc-tagged proteins were pulled down from CM derived from transfections as in (**A**), the amount of Fc-tagged protein from various CM normalized, and associated AP enzyme activity quantified, N = 3. Values for (**B**) and (**D**) are means ± SEM, *p<0.05, **p<0.01 by Student's t-test.

The online version of this article includes the following source data and figure supplement(s) for figure 7:

*Figure 7 continued on next page*

*Figure 7 continued*

**Source data 1.** Source data for quantification of AP activity in CDON-Fc pulldown experiments shown in *Figure 7B*.
**Source data 2.** Source data for quantification of AP activity in CDON-Fc and soluble LRP2 minireceptor-Fc pulldown experiments shown in *Figure 7D*.
**Figure supplement 1.** HEK293 cell transfectants expressing the indicated LRP2 mini-receptor were incubated with GST-SHH-N, and cell lysates immunoprecipitated with antibody to LRP2 and blotted with antibodies to LRP2 and GST.

in expression of Nodal target genes in the APS (including *Gsc*, *Foxa2*, and *Lefty2*), whereas EtOH treatment or *Cdon* mutation alone do not (*Hong and Krauss, 2012*; this study); (2) the window of sensitivity to EtOH-induced HPE is very narrow and closed by E7.5, a time reported to be prior to SHH expression and function in prechordal plate-mediated rostroventral midline patterning; (3) EtOH itself is the likely HPE-inducing teratogen and peak circulating EtOH levels coincide with Nodal signaling in the APS; (4) *Cdon* interacts genetically with the critical Nodal pathway factors *Tdgf1* and *Lefty2* in EtOH-induced HPE; (5) acute EtOH treatment dose-dependently diminishes p-SMAD2C and elevates p-SMAD2L levels, and inhibits Activin/Nodal-dependent gene expression, in mEpiSCs; (6) *Cdon;Lrp2* double mutants display phenotypes similar to mice with hypomorphic Nodal pathway mutations, including anterior truncations, a phenotype not seen in mice lacking all HH signaling; and (7) CDON and LRP2 both bind efficiently to TDGF1, an essential component of the Nodal receptor. We note that although EtOH inhibited p-SMAD2C production in mEpiSCs, and diminished expression of Nodal target genes in both mEpiSCs and *Cdon*[-/-] embryos, we have not yet demonstrated that EtOH inhibits p-SMAD2C production in embryos; additional work is therefore required to show the mechanisms in vitro and in vivo operate at the same level. Similarly, analysis of Nodal signaling in *Cdon;Lrp2* double mutant embryos requires study.

Together our findings are consistent with the following model: CDON and LRP2 function with overlapping or compensatory roles to regulate Nodal pathway signaling in the APS during induction of the PCP. Brief exposure of *Cdon*[-/-] embryos to EtOH during this period transiently and partially inhibits Nodal pathway activity. While all the mice successfully gastrulate, they emerge from this period predisposed towards development of HPE in a stochastic manner. If HH signaling strength fails to reach a required threshold early during rostroventral midline patterning, at the PCP stage, the outcome is alobar HPE and cyclopia; if it occurs at later stages of development, the outcomes are progressively less severe. Aspects of the model will require additional experimentation, but it is logically consistent with the synergistic effects of fetal alcohol and *Cdon* mutation within a narrow window of sensitivity, and also with the 'mutation plus modifier' view of human HPE. Our findings do not exclude that EtOH may also target HH signaling in *Cdon*[-/-] mice, but they are consistent with Nodal signaling being a major target of EtOH in HPE.

The specific mechanisms whereby acute EtOH treatment reduces expression of Nodal target genes in the APS of *Cdon*[-/-] mice and diminishes Activin/Nodal signaling in mEpiSCs are not known. EtOH itself, rather than consequences of its oxidative metabolism, is very likely the HPE-inducing teratogen (*Hong and Krauss, 2017*). An alternative mechanism to metabolism-based toxicity is that EtOH itself functions to perturb cell membranes via its hydrophobicity (*Lyon et al., 1981*; *McKarns et al., 1997*). One possibility is that EtOH's hydrophobic nature perturbs trafficking of Nodal receptor components or their stable assembly at the cell surface. TDGF1 is a GPI-linked protein, so might be especially vulnerable to hydrophobic membrane perturbation. However, exogenously provided Activin A is the major activator of SMAD2 in cultured mEpiSCs, and Activin does not require TDGF1 to signal through ActRII/ALK4. EtOH reduces p-SMAD2C levels in these cultures, so TDGF1 may not be a direct critical target of EtOH. Nevertheless, mEpiSCs endogenously express Nodal (*Kojima et al., 2014*), which does require TDGF1, and its contribution to SMAD2 signaling in these cultures is not fully clear. Additionally, EtOH activates JNK1 in mEpiSCs, with a corresponding increase in p-SMAD2L levels. Phosphorylation of SMAD2 by MAP kinases is generally inhibitory to SMAD2 activity, independent of p-SMAD2C phosphorylation status (*Massague, 2003*; *Rezaei et al., 2012*), so EtOH appears to promote or inhibit multiple signals that converge on SMAD2 phosphorylation.

CDON regulates several signaling pathways, but it has not previously been implicated in Nodal signaling. How CDON functions to regulate Nodal signaling is of obvious interest. *Cdon;Lrp2* double mutant mice display phenotypes that resemble Nodal pathway hypomorphs, and CDON and LRP2

each bind to TDGF1. Therefore, as with the effects of EtOH, TDGF1 is a possible point of regulation. One potential mechanism is that CDON and LRP2 might function as components of a larger Nodal receptor complex, contributing via interaction with TDGF1. Alternatively, they may work with TDGF1 in processing and trafficking of Nodal or other receptor components, or play a role in processing and trafficking of TDGF1 itself (*Constam, 2009*). There is evidence in other systems that CDON and LRP2 can act in such a manner. LRP2 is best understood as an endocytic receptor that controls internalization and intracellular trafficking of both soluble and membrane-bound proteins, including PTCH1, the $Na^+/H^+$ exchanger NHE3, and the intrinsic factor receptor, Cubilin (*Willnow and Christ, 2017*). Furthermore, CDON regulates subcellular localization of N-cadherin during neural crest migration in zebrafish (*Powell et al., 2015*). Similarly, the *Drosophila* orthologs of CDON act not only as co-receptors for HH, but in trafficking of the primary HH receptor, PTC (*Zheng et al., 2010*). Another possibility is in regulation of cell-cell contact and adhesion. CDON is localized to sites of cell-cell contact and functions with N-cadherin in cell adhesion-dependent signaling (*Kang et al., 2003*; *Lu and Krauss, 2010*). In zebrafish, a positive feedback loop exists between Nodal signaling, E-cadherin expression, and duration of cell-cell contact, that directs PCP specification at the expense of endoderm specification (*Barone et al., 2017*). It is conceivable that CDON could play a role in adhesion-mediated, higher order regulatory events like this as well. These varied potential mechanisms are not mutually exclusive.

Our findings offer insight into how *Cdon* mutation synergizes with fetal alcohol in mice to produce a wide spectrum of HPE phenotypes, closely resembling the complex etiology and variable outcomes seen in humans at the population level. A combination of approaches, including window of sensitivity studies, genetic interactions in mice, and in vitro analyses revealed that, unexpectedly, a major point of synergy is at the level of Nodal signaling. These findings illuminate gene-environment interactions in the causation of a common birth defect, expand understanding of alcohol teratogenesis, and serve as a conceptual framework for additional developmental anomalies.

# Materials and methods

## Key resources table

| Reagent type (species) or resource | Designation | Source or reference | Identifiers | Additional information |
|---|---|---|---|---|
| Genetic reagent (*Mus musculus*) | Cdon⁻ | MGI | MGI:1926387 | |
| Genetic reagent (*Mus musculus*) | Cripto⁻ | MGI | MGI:98658 | |
| Genetic reagent (*Mus musculus*) | Lefty2⁻ | MGI | MGI:2443573 | |
| Genetic reagent (*Mus musculus*) | Lrp2⁻ | MGI | MGI:95794 | |
| Cell line (Human) | HEK293T | ATCC | | |
| Cell line (*Mus musculus*) | EpiSC9 | *Najm et al., 2011* | | *Huang et al., 2017* *Kojima et al., 2014* |
| Antibody | anti-Smad2 (Rabbit mAB) | Cell Signaling | #5339 | WB (1:1000) |
| Antibody | anti-phospho-Smad2C (Rabbit mAB) | Cell Signaling | #3108 | WB (1:1000) |
| Antibody | anti-phospho-Smad2L (Rabbit mAB) | Cell Signaling | #3104 | WB (1:1000) |
| Antibody | anti-JNK (Rabbit polyclonal) | Cell Signaling | #9252 | WB (1:1000) |
| Antibody | anti-phospho-JNK (Rabbit polyclonal) | Cell Signaling | #9251 | WB (1:1000) |
| Antibody | anti-Gapdh (Mouse mAB) | Cell Signaling | #97166 | WB (1:5000) |

*Continued on next page*

*Continued*

| Reagent type (species) or resource | Designation | Source or reference | Identifiers | Additional information |
|---|---|---|---|---|
| Antibody | anti-DIG AP conjugated | Roche | 11093274910 | WM in situ (1:2000) |
| Antibody | anti-LRP2 (Goat) | *Willnow et al., 1996* | | WB (1:1000) |
| Recombinant DNA reagent | SHH-N-AP (plasmid) | This paper | | |
| Recombinant DNA reagent | CD164-AP (plasmid) | This paper | | |
| Recombinant DNA reagent | ActRIIA-AP (plasmid) | This paper | | |
| Recombinant DNA reagent | ALK4-AP (plasmid) | This paper | | |
| Recombinant DNA reagent | Cripto-AP (plasmid) | This paper | | |
| Recombinant DNA reagent | LRP2 sR1-Fc (plasmid) | This paper | | |
| Recombinant DNA reagent | LRP2 sR2-Fc (plasmid) | This paper | | |
| Recombinant DNA reagent | LRP2 sR3-Fc (plasmid) | This paper | | |
| Recombinant DNA reagent | LRP2 sR4-Fc (plasmid) | This paper | | |
| Recombinant DNA reagent | Cdon-Fc (plasmid) | *Kang et al., 2003* | | |
| Peptide, recombinant protein | Human Plasma Fibronectin purified protein | MilliporeSigma | FC010 | 10 µg/ml/cm$^2$ |
| Peptide, recombinant protein | Activin A | R and D Systems | 338-AC | 20 ng/ml |
| Peptide, recombinant protein | FGF2 | R and D Systems | 234-FSE | 12 ng/ml |
| Peptide, recombinant protein | IgG Fc (human) | Jackson Laboratories | 009-000-008 | |
| Peptide, recombinant protein | Protein G-agarose beads | Roche | 11243233001 | |
| Peptide, recombinant protein | Anti-AP-conjugated agarose beads | Sigma | A2080 | |
| Commercial assay or kit | AP yellow liquid substrate | Sigma | P7998 | |
| Commercial assay or kit | BM Purple | Roche | 11442074001 | |
| Commercial assay or kit | DIG-labeling kit | Roche | 11277073910 | |
| Commercial assay or kit | Effectene transfection reagent | Qiagen | 301425 | |
| Commercial assay or kit | RNA easy mini kit | Qiagen | 74104 | |
| Commercial assay or kit | Superscript III First strand synthesis system | Invitrogen | 18080051 | |

*Continued on next page*

*Continued*

| Reagent type (species) or resource | Designation | Source or reference | Identifiers | Additional information |
|---|---|---|---|---|
| Commercial assay or kit | iQ SyBR Green Supermix | BioRad | 1708882 | |
| Software, algorithm | Prism 8 | GraphPad | Prism 8 for MacOS ver 8.4.3 | |
| Sequence-based reagent | GAPDH_F | Invitrogen *Dong et al., 2008* | PCR primers | AACGACCCCTTCATTGAC |
| Sequence-based reagent | GAPDH_R | Invitrogen; *Dong et al., 2008* | PCR primers | TCCACGACATACTCAGCAC |
| Sequence-based reagent | Fgf5_F | Invitrogen; *Liu et al., 2018* | PCR primers | GCTGTGTCTCAGGGGATTGT |
| Sequence-based reagent | Fgf5_R | Invitrogen; *Liu et al., 2018* | PCR primers | CACTCTCGGCCTGTCTTTTC |
| Sequence-based reagent | Gbx2_F | Invitrogen; Harvard Primer Bank 133892275c2 | PCR primers | GCAACTTCGACAAAGCCGAG |
| Sequence-based reagent | Gbx2_R | Invitrogen; Harvard Primer Bank 133892275c2 | PCR primers | CCTTGCCCTTCGGGTCATC |
| Sequence-based reagent | Hoxa1_F | Invitrogen; *Matt et al., 2005* | PCR primers | CGCACAATGTTCTGATGTCC |
| Sequence-based reagent | Hoxa1_R | Invitrogen; *Matt et al., 2005* | PCR primers | TGCAAGCTTCATGACAGAGG |
| Sequence-based reagent | Lefty1_F | Invitrogen; *Liu et al., 2018* | PCR primers | AACCGCACTGCCCTTAT |
| Sequence-based reagent | Lefty1_R | Invitrogen; *Liu et al., 2018* | PCR primers | CGCGAAACGAACCAACTTGT |
| Sequence-based reagent | Lefty2_F | Invitrogen; *Liu et al., 2018* | PCR primers | CAGCCAGAATTTTCGAGAGGT |
| Sequence-based reagent | Lefty2_R | Invitrogen; *Liu et al., 2018* | PCR primers | CAGTGCGATTGGAGCCATC |
| Sequence-based reagent | Nanog_F | Invitrogen; *Chng et al., 2010* | PCR primers | GGACTTTCTGCAGCCTTACG |
| Sequence-based reagent | Nanog_R | Invitrogen; *Chng et al., 2010* | PCR primers | GCTTCCAAATTCACCTCCAA |
| Sequence-based reagent | Nodal_F | Invitrogen; *Liu et al., 2018* | PCR primers | CCTGGAGCGCATTTGGATG |
| Sequence-based reagent | Nodal_R | Invitrogen; *Liu et al., 2018* | PCR primers | ACTTTCTGCTCGACTGGACA |
| Sequence-based reagent | Pou5f1_F | Invitrogen; *Liu et al., 2018* | PCR primers | AGTTGGCGTGGAGACTTTGC |
| Sequence-based reagent | Pou5f1_R | Invitrogen; *Liu et al., 2018* | PCR primers | CAGGGCTTTCATGTCCTGG |
| Sequence-based reagent | Six1_F | Invitrogen; *Chng et al., 2010* | PCR primers | TTAAGAACCGGAGGCAAAGA |
| Sequence-based reagent | Six1_R | Invitrogen; *Chng et al., 2010* | PCR primers | GGGGGTGAGAACTCCTCTTC |
| Sequence-based reagent | Sox2_F | Invitrogen; *Liu et al., 2018* | PCR primers | GCGGAGTGGAAACTTTTGTCC |
| Sequence-based reagent | Sox2_R | Invitrogen; *Liu et al., 2018* | PCR primers | CGGGAAGCGTGTACTTATCCTT |
| Sequence-based reagent | T_F | Invitrogen; *Liu et al., 2018* | PCR primers | CTCGGATTCACATCGTGAGAG |
| Sequence-based reagent | T_R | Invitrogen; *Liu et al., 2018* | PCR primers | AAGGCTTTAGCAAATGGGTTGTA |

## Mice

All animal procedures were conducted in accordance with institutional guidelines for the care and use of laboratory animals as approved by the Institutional Animal Care and Use Committee (IACUC) of the Icahn School of Medicine at Mount Sinai and according to Max-Delbruck-Center guidelines following approval by local authorities (X9007/17). B6.129P2-$Lefty2^{tm1Hmd}$ mice were provided by Hiroshi Hamada. $Tdgf1$ mutant mice ($Cripto-LacZ$ mice) were provided by Michael Shen. Both strains were transferred onto the 129S6/SvEvTac (129S6) background with the Taconic Speed Congenic Program, by backcrossing and mapping with the 1450 SNP array. Mice used for these experiments were estimated to be over 98% 129S6/SvEvTac background. These lines were crossed with $Cdon^{+/tm1Rsk}$ ($Cdon^{+/-}$) mice on the 129S6 background (*Cole and Krauss, 2003*; *Hong and Krauss, 2012*) to generate 129S6.$Cdon^{+/-}$;$Lefty2^{+/-}$ and 129S6.$Cdon^{+/-}Tdgf1^{+/-}$ double mutant mice. Because $Cdon$ and $Tdgf1$ are both located on chromosome 9, $Cdon^{+/-}$;$Tdgf1^{+/-}$ mice were crossed with wild type mice and offspring screened for those that carried a recombinant chromosome 9 carrying both $Cdon$ and $Tdgf1$ mutant alleles. Offspring of intercrosses of these mice were further crossed (see *Tables 2* and *3* for details) and pregnant females treated with EtOH or saline as a control (*Hong and Krauss, 2012*). Briefly, two- to three-month old mice were mated for one hour in the dark and plugged females were collected. The time of the plug was designated as embryonic day (E) 0.0. Pregnant female mice were injected intraperitoneally twice with 15 µl per g body weight of solution of 30% EtOH in saline (3.48 g/kg) at E7.0 and again 4 hr later. For window-of-sensitivity experiments, pregnant females were treated similarly, except the initial dose was given at E7.25 or E7.5. For studies involving $Cdon$;$Tdgf1$ mutants, a lower dose of ethanol (25% EtOH in saline, 2.9 g/kg) was used. Saline injections were used as controls throughout.

Mice with targeted gene disruption of $Lrp2$ ($Lrp2^{tm1Her}$) have been described (*Willnow et al., 1996*). $Lrp2^{+/tm1Her}$ ($Lrp2^{+/-}$) mice were kept on the C57BL/6N genetic background and crossed with $Cdon^{+/-}$ mice on the C57BL/6N genetic background. Offspring were further intercrossed, and $Lrp2^{+/-}$;$Cdon^{+/-}$ mice were subjected to timed mating to collect embryos at the depicted embryonic day. Neither $Cdon^{-/-}$ mice nor $Lrp2^{-/-}$ mice on the C57BL/6N display anterior truncation phenotypes.

## Whole mount in situ hybridization

Whole mount RNA in situ hybridization was performed according to standard protocols (*Hong and Krauss, 2012*). Briefly, embryos were dissected out and fixed in 4% paraformaldehyde in phosphate-buffered saline (PBS), dehydrated through a graded methanol series, and stored at −20˚C. Rehydrated embryos were treated with 10 µg/ml proteinase K (Qiagen) in PBS, 0.1% Tween-20 (PBST) according to stage. Embryos were rinsed with PBST, post-fixed and hybridized with digoxygenin (DIG)-labeled probe in hybridization mix (50% formamide, 1.3x SSC pH 5.0, 5 mM EDTA, 50 µg/ml yeast tRNA, 100 µg/ml heparin, 0.2% Tween-20, and 0.5% CHAPS) overnight at 65˚C. They were washed, blocked with 2% Roche blocking reagent, 20% heat-inactivated lamb serum in Tris-buffered saline with 0.1% Tween-20 (TBST) and incubated with alkaline phosphate-conjugated anti-DIG antibody (1:2000, Roche) in blocking buffer overnight at 4˚C. After washing with TBST and NTMT (100 mM NaCl, 100 mM Tris pH9.5, 50 mM MgCl$_2$, and 0.1% Tween-20), embryos were stained with BM Purple AP substrate (Roche) in the dark. Stained embryos were cleared in 80% glycerol and photographed with a Jenoptik ProgRes C3 camera attached to Nikon SMZ 1500 stereomicroscope. Captured images were assembled by Helicon Focus software (Helicon Soft).

## Mouse epiblast stem cell culture and western blot and qRT-PCR analysis

Mouse epiblast stem cells (EpiSC9 cells) were obtained from Jianlong Wang, authenticated by robust expression of pluripotency markers, and confirmed to be negative for mycoplasma infection. EpiSC9 cells were cultured as described (*Huang et al., 2017*). Briefly, cells were cultured on 0.1% fibronectin-coated plates (MilliporeSigma, 10 µg/ml/cm$^2$) in N2B27 media supplemented with 20 ng/ml Activin A (R and D Systems) and 12 ng/ml Fgf2 (R and D Systems). Media were changed daily. EpiScs were treated with EtOH for six hours in fresh EpiSC media with supplements. For protein analysis, cells were harvested with RIPA buffer plus protease inhibitor (Sigma) and phosphatase inhibitor (Sigma). Western blotting was as described (*Bae et al., 2009*). Images were scanned and quantified using ImageJ software. Statistical significance was calculated using Student's t-test and a

cutoff of p<0.05. Primary antibodies used for western blot: SMAD2 (Cell Signaling #5339), p-SMAD2C (Cell Signaling #3108), p-SMAD2L (Cell Signaling #3104), JNK (Cell Signaling #9252), p-JNK (Cell Signaling #9251), and GAPDH (Cell Signaling #97166). For mRNA analysis, total RNA was extracted from EpiSCs using the RNAeasy kit (Quiagen). Reverse transcription and cDNA production were performed with Superscript III first strand synthesis system (Invitrogen). qPCR was performed using iQ SyBR green supermix (BioRad) on an iCycler iQ5 (BioRad). Gene expression levels were normalized to *Gapdh*.

## Protein-protein interactions

Expression vectors encoding soluble, tagged forms of CDON, SHH-N, CD164, ActRIIA, ActRIIB, ALK4, TDGF1, and LRP2, and LRP2 mini-receptors that encoded the LRP2 transmembrane and intracellular domains, were constructed by standard methods; details are available on request. HEK293T cells were obtained from ATCC, confirmed to be negative for mycoplasma infection, and cultured in DMEM with 10% FBS and 1x Penicillin/Streptomycin. HEK293T cells in 10 cm dishes were transfected with a total of either 2 or 3 µg of plasmid vectors using Effectene reagent (Qiagen). Media were changed to 2% FBS two days after transfection. Conditioned media were harvested five days afterwards. Soluble fusion proteins in conditioned media were quantified by dot blotting with human IgG Fc protein (Jackson ImmunoResearch Laboratories) as a standard, and by alkaline phosphatase (AP) enzyme activity using AP yellow liquid substrate (Sigma). For analysis of CDON-Fc interactions with soluble AP-tagged proteins, the factors were coexpressed in HEK293T cells. For Fc pull down and AP activity assays with CDON-Fc, soluble fusion proteins were incubated with protein G agarose beads (Roche) for 2 hr at 4°C. For analysis of soluble LRP2-Fc mini-receptors interactions with soluble AP-tagged proteins, LRP2 sR1-sR4 and the AP-tagged counterparts were generated separately in individual transfections and CM mixed together. Pull down complexes were washed five times with ice-cold HNTG buffer (20 mM HEPES pH 7.5, 150 mM NaCl, 0.1% Triton X-100, 10% glycerol). Endogenous AP was inactivated by heat inactivation. AP activity in pull-down complexes was measured using AP yellow liquid substrate (Sigma). For AP pull down followed by western blot analysis, soluble fusion proteins were incubated with anti-AP conjugated agarose beads (Sigma) overnight at 4°C. Pull-down complexes were washed five times with HNTG buffer, eluted from the beads by boiling, and separated by SDS-PAGE. Antibodies used for western blot were HRP conjugated with goat anti-human Fc.

To test SHH-N interaction with membrane-bound LRP2 mini-receptors, NIH3T3 cells were transfected with the four individual LRP2 mini-receptors encoding transmembrane and intracellular domains and incubated with 20 µg/ml GST-SHH-N in the medium for two hours. Cell lysates were immunoprecipitated with polyclonal LRP2 antibody directed against full-length LRP2, and subjected to western blot analysis with anti-LRP2 and anti-GST antibodies.

## Acknowledgements

We thank Horoshi Hamada for providing *Lefty2* mutant mice and reagents; Michael Shen for providing *Tdgf1* mutant mice and reagents; Zoe Fresquez and Marysia-Kolbe Rieder for technical support; Xin Huang and Jianlong Wang for EpiSC9 cells and advice; Nadeera Wickramasinghe and Nicole Dubois for generous support with cell culture; Steve Farber and Phil Soriano for helpful discussions, and members of the Krauss lab for reading the manuscript.

## Additional information

### Funding

| Funder | Grant reference number | Author |
| --- | --- | --- |
| National Institute of Dental and Craniofacial Research | DE024748 | Robert S Krauss |
| German Research Foundation | CH1838/1-1 | Annabel Christ |

The funders had no role in study design, data collection and interpretation, or the decision to submit the work for publication.

## Author contributions

Mingi Hong, Conceptualization, Validation, Investigation, Visualization, Methodology, Writing - review and editing; Annabel Christ, Conceptualization, Funding acquisition, Validation, Investigation, Visualization, Methodology, Writing - review and editing; Anna Christa, Validation, Investigation, Visualization, Methodology, Writing - review and editing; Thomas E Willnow, Conceptualization, Resources, Supervision, Funding acquisition, Project administration, Writing - review and editing; Robert S Krauss, Conceptualization, Resources, Supervision, Funding acquisition, Writing - original draft, Project administration

## Author ORCIDs

Robert S Krauss (iD) https://orcid.org/0000-0002-7661-3335

## Ethics

Animal experimentation: All animal procedures were conducted in accordance with institutional guidelines for the care and use of laboratory animals as approved by the Institutional Animal Care and Use Committee (IACUC) of the Icahn School of Medicine at Mount Sinai (protocol number 14-0191) and according to Max-Delbruck-Center guidelines following approval by local authorities (protocol number X9007/17).

## Decision letter and Author response

Decision letter https://doi.org/10.7554/eLife.60351.sa1
Author response https://doi.org/10.7554/eLife.60351.sa2

# Additional files

## Supplementary files

• Transparent reporting form

## Data availability

All data generated or analysed during this study are included in the manuscript and supporting files.

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
