## [Decision Letter]

**Acceptance summary:**

The manuscript dissects the interaction between alcohol teratogenesis and *Cdon* mutations as causes of brain and face malformations typical of a rare birth defect known as holoprosencephaly. The manuscript shows that mechanistically the two factors converge in a very restricted temporal window on the Nodal signaling pathway, providing novel insight on how an environmental factor can impinge upon an otherwise silent genetic alteration.

**Decision letter after peer review:**

Thank you for submitting your article "*Cdon* mutation and fetal alcohol converge on Nodal signaling in a mouse model of holoprosencephaly" for consideration by *eLife*. Your article has been reviewed by Marianne Bronner as the Senior Editor, a Reviewing Editor, and three reviewers. The following individuals involved in review of your submission have agreed to reveal their identity: Robert Lipinski (Reviewer #1); Paul Kruszka (Reviewer #2).

The reviewers have discussed the reviews with one another and the Reviewing Editor has drafted this decision to help you prepare a revised submission.

Summary:

Holoprosencephaly (HPE) is a common birth defect in human with genetic and environmental causes. Prominent among the genetic causes are component of the Nodal and Hedgehog signaling pathways. In previous studies the group of R. Krauss has shown that mutations in the *Cdon* gene (which is known to act as a HH binding protein) and in utero exposure to ethanol produced HPE with a penetrance not observed when the two factors are considered individually. The present study offers an important mechanistic insight into mechanism underlying this synergy. It shows that Nodal is the target upon which prenatal alcohol exposure converges with *Cdon* mutations in causing the brain and face malformations of holoprosencephaly. The manuscript thus expands on the present understanding of alcohol teratogenesis and *Cdon* role in embryogenesis demonstrating how these factors interact.

Essential revisions:

There is general consensus that this a very valuable and well performed study that adds important information on the interaction between environmental and genetic factors in the development of HPE.

There are however a number of issues that need your attention:

1) The data rule out the participation of HH signaling in the presented phenotype are not strong and mostly based on previous studies in which differences in mouse strain, breeding etc. may make the comparison with the present study questionable. Thus, at minimum, you reconsider the interpretation of the data and tone down the conclusions.

2) There is also too much speculation related to the role of CDON variants in humans. This needs to be addressed.

3) Please also address the state of p-SMAD2C in ethanol treated embryos and in *Cdon*, *Lrp2* double mutant embryos.

4) To sustain the conclusions related to the Nodal signaling activity please also analyze additional target genes such as *Gsc*.

Reviewer #1:

General assessment: The manuscript by Hong et al. provides evidence pointing toward Nodal as a target upon which prenatal alcohol exposure converges with *Cdon* mutations in causing the brain and face malformations of holoprosencephaly. Dissecting the interaction of specific genetic and chemical influences is an important but challenging focus of birth defects research. The manuscript offers an exceptional degree of mechanistic insight that challenges the existing understanding of alcohol teratogenesis, expands the role of *Cdon* in embryogenesis, and points to how these factors interact. While some interpretations rely on interpolation of historical data and probably should be dialled back, on balance the study appears high quality and the findings novel, making the work as a whole a significant advance in the field.

1) The data supporting ethanol acting (at least in part) upstream of HH on Nodal is strong and compelling. However, the argument against alcohol acting (at least in part) on HH pathway is less convincing, not supported by strong direct evidence, and relies on comparison of new and historical data. For example, the authors state that the window of sensitivity to EtOH-induced HPE in 129S6 *Cdon*^-/-^ mice is closed by E7.5, prior to *Shh* expression in the midline which begins at E7.5-7.75 but contemporaneous evidence is lacking. The strain of mice, breeding strategy, and staging nomenclature may have differed between the present study and that cited for *Shh* expression timing (Echalard). The authors show no effect of ethanol exposure at E7.5. Dosing at E7.25 however does have an effect but this follows a dose at E7.25 and another 4 hours later, making it likely that ethanol exposure persists through E7.5. Dialling back the interpretation of these data and focusing more on those supporting Nodal as a newly identified target of ethanol and point of synergism between ethanol and *Cdon* would mitigate this concern. This need not detract from the impact of the novel findings presented. As the authors note, alcohol may perturb cell membranes and receptor trafficking/function, so mechanisms/targets are unlikely to be mutually exclusive.

Reviewer #2:

This is a very well written manuscript and important for HPE research. I believe this is a "must publish" manuscript. This study addresses the "Holy Grail" of genetics of modifiers, incomplete penetrance and variable expressivity. This study is also a beautiful continuation of this group's narrative on CDON and ETOH.

Until this study, CDON was thought to act mainly through HH signaling, and this is a first in showing the CDON-Nodal signaling relationship. The mouse knockout and epiblast stem cell experiments are fantastic and convincing. There few studies with this high-quality figures.

My most significant comment is that the authors speculate too much about the role of CDON variants in humans. If you dig deep into the published literature, there is not a strong association with CDON and HPE in humans (driver or modifier). There is one fact that cannot be ignored: in healthy humans, loss of function variants in CDON is common, which is very different from the known associated HPE genes such as SHH, ZIC2, SIX3, STAG2, SMC1A etc. Therefore, it will take a large epidemiologic study to connect CDON and ETOH in humans. The data is just not there in humans to connect alcohol and CDON variation (yet).

1) Introduction: "animal models are required…" is debatable. The mouse model behaves very differently than humans in regard to the genetics of HPE; for example, only a heterozygous variant in SHH results in HPE in humans where heterozygous mice with SHH variants are normal. Animal models are a surrogate for human studies that are underpowered; especially, in conditions such as HPE which is rare. There have been attempts in humans to answer the question of environmental effects on HPE, see Addissie et al., 2020, again study power is an issue.

2) Introduction: Addissie et al., 2020 should be referenced here. Also, recommend reviewing Addissie et al.

3) I do not believe there is sufficient evidence to associate CDON with HPE in humans. In the largest study mentioning CDON's association with HPE, some of the variants were (1) transmitted from a parent; and (2) a missense variant without functional evidence of pathogenicity. These CDON variants are variants of unknown significance. Commercial labs often misinterpret these gene associations with HPE and it is very important to be precise here, i.e. see https://www.invitae.com/en/physician/tests/04424/#info-panel-assay_information. See my next comment. I believe the authors have misinterpreted the findings in Roessler et al., 2018a.

4) It should be noted that CDON loss of function variants are commonly found in healthy human individuals (observed / expected ration of 0.58; https://gnomad.broadinstitute.org/gene/ENSG00000064309?dataset=gnomad_r2_1), showing that CDON loss of function variants are not constrained by evolution. This is very important and it supports the authors' findings that most alcohol is needed with a CDON variant to reach the HPE threshold in mice and most likely (though not proven) in humans.

Reviewer #3:

Holoprosencephaly (HPE) is a common birth defect in human. Many genes such as those in Nodal and Hedgehog signaling are implicated in HPE, but environmental factors also contribute. The authors previously showed that a mutation in *Cdon* gene or in utero exposure to ethanol individually had little effect, but the combination of both produced HPE with high penetrance. However, it was unknown how *Cdon* mutation and ethanol synergize for HPE.

In this paper, the authors conclude that *Cdon* and ethanol converge to Nodal signaling. Involvement of *Cdon* in Nodal signaling was supported by genetic and physical interaction between *Cdon* and Cripto (a coreceptor of Nodal). Effects of ethanol on Nodal signaling was examined with mouse epiblast stem cells, and ethanol was shown to inhibit C-terminal phosphorylation of SMAD2, a readout of Nodal activity. Collectively, it was concluded that *Cdon* mutation and in utero ethanol synergistically inhibit Nodal signaling, producing HPE.

In all, this paper provides a nice example showing how interaction between a gene and an environmental factor affects a particular developmental process. Since many of genetic disorders in human likely involve multiple factors, genetic and environmental factors, this paper has important messages. To be published in *eLife*, however, the following points need to be addressed.

1) Authors provide a nice set of data showing that ethanol inhibits SMAD2 C-terminal phosphorylation in mouse epiblast stem cells (Figure 2A, 2B). This is excellent. However, it is more important to see the effects of ethanol with mouse embryos. This can be done by staining of mouse embryos such as those shown in Figure 1 with the anti-phosphorylated SMAD2C (the antibody is commercially available and has been used by others). Similarly, I would like to suggest examining p-SMAD2C in *Cdon*, *Lrp2* double mutant embryos too (Figure 3A; whole mount staining may be easier at an earlier stage than E11.5). This would strengthen the conclusion that *Cdon* and ethanol synergistically affect Nodal signaling in mouse embryos.

2) *Lefty2* expression is regulated by Nodal, but *Lefty2* protein inhibits Nodal activity, which makes a complex network. It is preferable to examine additional Nodal target genes such as Gsc in Figure 1. Also, is Nodal expression maintained in *Cdon*^-/-^, EtOH embryos (Figure 1)?

---

## [Author Response]

Essential revisions:There is general consensus that this a very valuable and well performed study that adds important information on the interaction between environmental and genetic factors in the development of HPE.There are however a number of issues that need your attention:1) The data rule out the participation of HH signaling in the presented phenotype are not strong and mostly based on previous studies in which differences in mouse strain, breeding etc. may make the comparison with the present study questionable. Thus, at minimum, you reconsider the interpretation of the data and tone down the conclusions.

We agree with the reviewer’s point and have modified the text in the Abstract, Results section, and Discussion section to temper our conclusions. Additionally, we state in the Discussion section, “Our findings do not exclude that EtOH may also target HH signaling in *Cdon*^-/-^ mice, but they are consistent with Nodal signaling being a major target of EtOH in HPE.”

2) There is also too much speculation related to the role of CDON variants in humans. This needs to be addressed.

We agree that the role of CDON variants in human HPE is not fully clear. We have made this overt in the Introduction.

3) Please also address the state of p-SMAD2C in ethanol treated embryos and in Cdon, Lrp2 double mutant embryos.

As described above, we will perform these experiments as soon as possible and report them as suggested by the editors. We have added text to the Discussion section to state explicitly that these experiments are required.

4) To sustain the conclusions related to the Nodal signaling activity please also analyze additional target genes such as Gsc.

We thank the reviewer for this comment. We have in fact analyzed expression of *Gsc* and *FoxA2*, two additional Nodal target genes important for PCP induction. Expression of *Gsc* and *FoxA2* are specifically reduced in EtOH-treated *Cdon* mutant embryos, similar to expression of *Lefty2*. These results were published in Hong and Krauss (2012), cited in the references. This was stated in the original version of the paper, but we have made this clearer in the revised manuscript (subsection *“Cdon* interacts genetically with Nodal pathway components” and Discussion section).

Reviewer #1:[…]1) The data supporting ethanol acting (at least in part) upstream of HH on Nodal is strong and compelling. However, the argument against alcohol acting (at least in part) on HH pathway is less convincing, not supported by strong direct evidence, and relies on comparison of new and historical data. For example, the authors state that the window of sensitivity to EtOH-induced HPE in 129S6 Cdon^-/-^ mice is closed by E7.5, prior to Shh expression in the midline which begins at E7.5-7.75 but contemporaneous evidence is lacking. The strain of mice, breeding strategy, and staging nomenclature may have differed between the present study and that cited for Shh expression timing (Echalard). The authors show no effect of ethanol exposure at E7.5. Dosing at E7.25 however does have an effect but this follows a dose at E7.25 and another 4 hours later, making it likely that ethanol exposure persists through E7.5. Dialling back the interpretation of these data and focusing more on those supporting Nodal as a newly identified target of ethanol and point of synergism between ethanol and Cdon would mitigate this concern. This need not detract from the impact of the novel findings presented. As the authors note, alcohol may perturb cell membranes and receptor trafficking/function, so mechanisms/targets are unlikely to be mutually exclusive.

We have made the requested changes to the text (please see the response to Essential revisions, point 1, above).

Reviewer #2:[…]My most significant comment is that the authors speculate too much about the role of CDON variants in humans. If you dig deep into the published literature, there is not a strong association with CDON and HPE in humans (driver or modifier). There is one fact that cannot be ignored: in healthy humans, loss of function variants in CDON is common, which is very different from the known associated HPE genes such as SHH, ZIC2, SIX3, STAG2, SMC1A etc. Therefore, it will take a large epidemiologic study to connect CDON and ETOH in humans. The data is just not there in humans to connect alcohol and CDON variation (yet).

We have made the requested changes to the text (please see the response to Essential revisions, point 2, above).

1) Introduction: "animal models are required…" is debatable. The mouse model behaves very differently than humans in regard to the genetics of HPE; for example, only a heterozygous variant in SHH results in HPE in humans where heterozygous mice with SHH variants are normal. Animal models are a surrogate for human studies that are underpowered; especially, in conditions such as HPE which is rare. There have been attempts in humans to answer the question of environmental effects on HPE, see Addissie et al., 2020, again study power is an issue.

We thank the reviewer for this comment. We have substituted the word “valuable” for the word “required”. We point out that this statement is consistent with a statement from HPE epidemiologists: “More data from in vitro and animal models might be necessary to identify the most biologically plausible risk factors.” (Miller at al. 2010, in the references).

We agree the Addissie et al., paper is highly relevant to the discussion and have cited it twice in the text (Results section).

2) Introduction: Addissie et al., 2020 should be referenced here. Also, recommend reviewing Addissie et al.

We agree the Addissie et al. paper is highly relevant to the discussion and have cited it twice in the text (Results section).

3) I do not believe there is sufficient evidence to associate CDON with HPE in humans. In the largest study mentioning CDON's association with HPE, some of the variants were (1) transmitted from a parent; and (2) a missense variant without functional evidence of pathogenicity. These CDON variants are variants of unknown significance. Commercial labs often misinterpret these gene associations with HPE and it is very important to be precise here, i.e. see https://www.invitae.com/en/physician/tests/04424/#info-panel-assay_information. See my next comment. I believe the author have misinterpreted the findings in Roessler, 2018a.4) It should be noted that CDON loss of function variants are commonly found in healthy human individuals (observed / expected ration of 0.58; https://gnomad.broadinstitute.org/gene/ENSG00000064309?dataset=gnomad_r2_1), showing that CDON loss of function variants are not constrained by evolution. This is very important and it supports the authors' findings that most alcohol is needed with a CDON variant to reach the HPE threshold in mice and most likely (though not proven) in humans.

We have made the requested changes to the text (please see the response to Essential revisions, point 2, above).

Reviewer #3:[…]1) Authors provide a nice set of data showing that ethanol inhibits SMAD2 C-terminal phosphorylation in mouse epiblast stem cells (Figure 2A, 2B). This is excellent. However, it is more important to see the effects of ethanol with mouse embryos. This can be done by staining of mouse embryos such as those shown in Figure 1 with the anti-phosphorylated SMAD2C (the antibody is commercially available and has been used by others). Similarly, I would like to suggest to examine p-SMAD2C in Cdon, Lrp2 double mutants embryos too (Figure 3A; whole mount staining may be easier at an earlier stage than E11.5). This would strengthen the conclusion that Cdon and ethanol synergistically affect Nodal signaling in mouse embryos.

In lieu of not performing these experiments immediately, we have made requested changes to the text (please see the response to Essential revisions, point 3, above). We will perform these experiments and report back as soon as possible.

2) Lefty2 expression is regulated by Nodal, but Lefty2 protein inhibits Nodal activity, which makes a complex network. It is preferable to examine additional Nodal target genes such as Gsc in Figure 1. Also, is Nodal expression maintained in Cdon^-/-^, EtOH embryos (Figure 1)?

We thank the reviewer for this comment. We have indeed assessed expression of *Gsc* and *FoxA2*. This was mentioned in the first version of the manuscript, but we have now made this clearer (please see the response to Essential revisions, point 4, above). We will assess Nodal expression at the same time we examine p-SMAD2C production in embryos and report back.